# Association of zinc deficiency with infectious complications in pediatric hematopoietic stem cell transplantation patients

**Warangkhana Suwanphoerung**[1], **Chompunuch Klinmalai**[1], **Sasivimol Rattanasiri**[2], **Samart Pakakasama**[1], **Usanarat Anurathapan**[1], **Suradej Hongeng**[1], **Nalinee Chongviriyaphan**[1], **Nopporn Apiwattanakul**[1]*

1 Department of Pediatrics, Faculty of Medicine Ramathibodi Hospital, Mahidol University, Bangkok, Thailand, 2 Section for Clinical Epidemiology and Biostatistics, Faculty of Medicine Ramathibodi Hospital, Mahidol University, Bangkok, Thailand

* nopporn.api@mahidol.ac.th

## Abstract

### Background

Zinc plays essential roles in immune function and epithelial integrity. Patients undergoing hematopoietic stem cell transplantation (HSCT) often have low plasma zinc levels because of poor intake and diarrhea. We hypothesized that patients with zinc deficiency before HSCT had worse infectious complications after HSCT compared with patients with normal zinc levels. Citrulline, a marker of intestinal integrity, was also hypothesized to be lower in patients with zinc deficiency.

### Patients and methods

Thirty patients undergoing HSCT at Ramathibodi Hospital during March 2020–September 2021 were enrolled. Blood samples for plasma zinc and citrulline were collected during the HSCT period. The 14- and 90-day outcomes after HSCT were prospectively recorded.

### Results

Twelve of 30 (40%) patients had zinc deficiency before HSCT. Zinc-deficient patients were younger (median (interquartile range): 6 (8.8) vs 13 (5.8) years old; $p = 0.017$). Zinc levels tended to increase after admission in both groups. Patients with zinc deficiency had lower citrulline levels than those with normal zinc levels. Citrulline levels decreased in both groups after stem cell infusion, and the level was not significantly different between the two groups. Zinc-deficient patients had a higher rate of bacterial infection within 90 days after HSCT than those with normal zinc levels (6 in 12 patients (50.0%) vs 1 in 18 patients (5.6%); odds ratio [OR]: 17.0; 95% confidence interval [CI]: 1.68–171.70; $p = 0.016$). This remained significant after adjustments for age (adjusted OR: 12.31; 95% CI: 1.084–139.92; $p = 0.043$).

**Data Availability Statement:** All relevant data are within the paper and its Supporting information files.

**Funding:** No.

**Competing interests:** The authors have declared that no competing interests exist.

**Abbreviations:** CI, confidence interval; ELISA, enzyme-linked immunosorbent assay; GVHD, Graft versus host disease; HSCT, hematopoietic stem cell transplantation; ICU, intensive care unit; OR, odds ratio; SD, standard deviation; VOD, veno-occlusive disease.

## Conclusion

The prevalence of zinc deficiency in pediatric patients undergoing HSCT was high. Zinc-deficient patients had lower citrulline levels and higher incidence of bacterial infection after HSCT. However, citrulline level was not different between patients with and without bacterial infections. It is worth to investigate whether zinc supplementation before HSCT can reduce bacterial infection after HSCT.

## Introduction

Zinc is an essential mineral for immune function. It regulates white blood cell chemotaxis and phagocytosis [1] and contributes to normal T cell lymphocyte function and consequently normal antibody production [2]. It also plays a role in skin and intestinal epithelial integrity by strengthening tight junctions in the epithelium [3, 4]. Zinc reduces proinflammatory cells in the gut, resulting in decreased gut epithelial inflammation and a subsequent reduction in epithelial leakage [5, 6]. Zinc deficiency may increase the risk of infection and other complications related to impaired mucosal barrier function [7], leading to continuous exposure to pathogens [8]. Malnourished children or children with critical illnesses have a higher probability of zinc deficiency, leading to severe hospital-acquired infections, increased morbidity, and longer hospitalization [9, 10].

Hematopoietic stem cell transplantation (HSCT) has been widely performed to treat oncologic, hematologic, immunologic, and hereditary diseases [11]. During HSCT, patients receive high doses of chemotherapy followed by stem cell infusions [12]. Patients undergoing HSCT are usually susceptible to chemotherapy-induced impaired mucosal barrier function and infections [13]. These patients might also be predisposed to low plasma zinc levels because of poor intake, decreased zinc absorption due to mucositis, and increased zinc loss resulting from diarrhea [14]. Because zinc is important for both an intact immune system and epithelial integrity, we hypothesized that zinc-deficient HSCT patients have worse infectious complications compared with patients with normal zinc levels. Given that citrulline, an amino acid derived from enterocytes in the small intestine, has been used as a marker of intestinal integrity and is strongly correlated with small intestinal enterocyte loss and mucositis [15–17], we also aimed to investigate whether the citrulline level is lower in zinc-deficient patients.

## Methods

### Study design and patients

Pediatric patients aged 0–18 years old who underwent HSCT at Ramathibodi Hospital in Thailand during March 2020–September 2021 were enrolled in this study. Patients who received a second transplantation or zinc supplementation before HSCT were excluded. Patients' demographic data, including age at stem cell transplantation, sex, underlying diseases, concurrent medications, weight, height, and requirement of parenteral nutrition, were recorded. The primary outcome of interest was microbiologically confirmed bacterial infection within 14 and 90 days after stem cell infusion. Bacterial infection was defined as the presence of bacteremia or localized organ infection with positive cultures. Bacteremia was defined as the presence of bacteria in at least one blood culture specimen. If the pathogens isolated were skin flora, the pathogens had to be isolated from at least two blood culture specimens. The secondary outcomes included viral infections, fungal infections, the incidence of fever after stem cell infusion

within 48 hours, engraftment syndrome [18, 19], mucositis within 90 days, veno-occlusive disease (VOD) [20] within 90 days, acute graft-versus-host disease (GVHD) [21] within 90 days, primary graft failure [22], intensive care unit (ICU) admission, and mortality within 90 days after HSCT.

After admission, all patients were provided a regular bacteria-free diet. They received a pre-transplant conditioning regimen, followed by stem cell infusion. GVHD prophylaxis was given after stem cell infusion. Acyclovir, ciprofloxacin, itraconazole, and penicillin V were given to prevent infection in all patients until neutrophil engraftment. Patients with invasive pulmonary aspergillosis still undergoing treatment were given voriconazole instead of itraconazole.

Patients who received haploidentical stem cell transplantation received stem cells from their father or mother. Those who received matched related donor stem cell transplantation received stem cells from their siblings and those who received matched unrelated donor stem cell transplantation received stem cells from Thai Red Cross. Thai Red Cross is a non-profit-able organization which manages to provide stem cells from registered donors for patients who need stem cell transplantation but do not have matched related donors. All donors or legally authorized representatives in haploidentical and matched related stem cell transplantation provided written informed consent that was freely given. Cash payments or any incentives were not offered to the family of the donors.

Morning blood samples were collected for serum zinc level analysis on the day of admission, day of stem cell infusion (day 0), and day+7, +14, and +28 after stem cell infusion. Serum citrulline was evaluated on the day of admission and day+7 after stem cell infusion. Plasma samples were stored at −20˚C until analysis. Zinc and citrulline levels were blinded throughout the study period. Zinc levels were measured with the standard Frame Atomic Absorption Spectrophotometry method. The sensitivity level of detection was 20 μg/dL, and the maximum level of detection was 200 μg/dL. The reference range for normal levels was 70–120 μg/dL. Patients with serum zinc levels <70 μg/mL at admission were classified as the zinc deficiency group, whereas those with serum zinc levels ≥70 μg/mL were classified as the normal group. The serum citrulline level was determined using ethylenediaminetetraacetic acid-treated blood with the Human Citrulline ELISA Kit from MyBioSource incorporation (San Diego, CA, USA). The technique was based on citrulline-antibody interactions, and a horseradish peroxidase colorimetric detection system was used for detection. The kit was designed to detect only native citrulline, not recombinant citrulline. The sensitivity of detection was 0.5 mmol/L, and citrulline deficiency was defined as serum citrulline levels below 20 mmol/L. The sensitivity and specificity were 80% and 84%, respectively [16]. Normal healthy children had a median plasma citrulline level of 34.36 mmol/L [23].

The study protocol was approved by the ethics committee of the Faculty of Medicine Ramathibodi Hospital, Mahidol University (MURA2020/682). Patients or legally authorized representatives gave written informed consent. Patients aged 7–18 years old also gave written assent form.

## Statistical analysis

All statistical analyses were performed using the STATA program version 17 (STATA Corp, College Station, TX, USA). The normal distribution of continuous variables was determined by the Shapiro–Wilk test. Continuous data with a normal distribution were summarized as the mean ± standard deviation (SD), and data not normally distributed were shown as the median (interquartile range). The comparisons of zinc and citrulline levels between groups or different time points were performed by mixed effects linear regression. Comparisons of other continuous data between groups were conducted by Student's t-test for normally distributed data and

Mann–Whitney U test for non-normally distributed data. The comparisons of categorical variables were performed by chi-square test. A p-value < 0.05 was considered statistically significant.

## Results

Thirty-one pediatric patients underwent HSCT during March 2020–September 2021. One patient who received a second HSCT was excluded. Thirty patients aged 1.0–17.8 years were included in this study. Patients' characteristics are summarized in Table 1. Patients with underlying malignancies included those diagnosed with acute lymphoblastic leukemia, chronic myeloid leukemia, neuroblastoma, Langerhans cell histiocytosis, medulloblastoma, and Hodgkin's lymphoma. Non-malignant diseases included thalassemia, bone marrow failure, osteopetrosis, chronic granulomatous disease, and multiple sclerosis. All patients had full enteral feeding before admission. No patients had signs of zinc deficiency, dermatitis, rash, or mucositis at admission. Zinc deficiency was found in 12 patients (40%; 95% confidence interval [CI]: 22.7–59.4%). The mean (±SD) zinc level at admission in patients with a normal zinc level was 81.89 ± 7.38 μg/dL, whereas that in patients with zinc deficiency was 59.67 ± 8.32 μg/dL. In both groups, the zinc level was lowest on admission day, and the level increased during HSCT (Fig 1).

In the zinc deficiency group, the zinc level tended to increase after HSCT and was significantly higher on day 0 ($p = 0.015$) and day+14 ($p = 0.003$) than that on admission day (Fig 1). Factors associated with zinc deficiency were presented in Table 1. Zinc-deficient patients were younger, had a shorter duration of primary diseases before HSCT, and were less likely to have undergone iron chelation therapy. Sex, height, and weight for age Z-scores, zinc supplementation during HSCT, and HSCT types were not different between the groups.

The citrulline level at admission was lower in zinc-deficient patients. The mean (±SD) citrulline level at admission was 30.8 ± 7.6 mmol/L in patients with normal zinc levels and 25.68 ± 5.78 mmol/L ($p = 0.034$) in patients with zinc deficiency. The citrulline level at day+7 was lower than that at admission in both groups, but this was not significantly different between the two groups at day+7 (Fig 2).

### Post-HSCT non-infectious outcomes

The incidences of febrile episodes within 48 hours after HSCT not attributed to infection, primary graft failure, engraftment syndrome, mucositis, VOD, acute GVHD, severe acute GVHD (grade 3 and 4), ICU visit, and death were not different between patients with zinc deficiency and patients with normal zinc levels before HSCT. Maximum body temperature within 48 hours after HSCT and days of white blood cell and platelet engraftments were also not different between the two groups. These results are summarized in Table 2.

### Post-HSCT infectious outcomes

As shown in Table 3, the incidences of febrile neutropenia, viral infection, viral hemorrhagic cystitis, and fungal infection within 90 days after HSCT were not different between patients with normal zinc or low zinc level before HSCT. Interestingly, patients with zinc deficiency had higher incidences of microbiologically confirmed bacterial infection within 90 days after HSCT ($p = 0.009$) and bacteremia within 90 days after HSCT ($p = 0.006$). Factors associated with bacterial infection in this cohort is summarized in Table 4. Patients with bacterial infection were younger and had lower zinc level before HSCT. Patients with zinc deficiency before HSCT were 17 times (95% CI: 1.68–171.70; $p = 0.016$) more likely to have bacterial infection within 90 days after HSCT. This likelihood was 12.3 times (95% CI: 1.08–139.92; $p = 0.043$)

**Table 1. Demographic data of enrolled patients.**

| | Total (N = 30) | Normal zinc group[*] (N = 18) | Zinc deficiency group[*] (N = 12) | p-value |
|---|---|---|---|---|
| Age (years), (IQR) | 10.6 (9.8) | 13 (5.8) | 6 (8.8) | **0.017** |
| Sex N (%) | | | | |
| • Male | 16 (53.3) | 11 (61.1) | 5 (41.7) | 0.296 |
| • Female | 14 (46.7) | 7 (38.9) | 7 (58.3) | |
| Primary disease N (%) | | | | |
| • Malignancy | 14 (46.7) | 9 (50.0) | 5 (41.7) | 0.722 |
| • Non-malignancy | 16 (53.3) | 9 (50.0) | 7 (58.3) | |
| Duration of primary disease before HSCT (years) (IQR) | 4.98 (10.0) | 8.73 (9.0) | 2.04 (4.0) | **0.010** |
| Height for age Z score N (%) | | | | |
| • Normal | 25 (83.3) | 12 (66.7) | 9 (75.0) | 0.364 |
| • Stunting | 5 (16.7) | 2 (11.0) | 3 (25.0) | |
| Weight for age Z score N (%) | | | | |
| • Normal | 30 (100) | 18 (100) | 12 (100) | NA |
| • Underweight | 0 (0) | 0 (0) | 0 (0) | |
| Total parenteral nutrition with zinc supplementation after HSCT N (%) | 17 (60.7) | 10 (55.6) | 7 (58.3) | 1.000 |
| Dose of zinc supplementation in total parenteral nutrition (mg/kg/day) (IQR) | 0.9 (0.2) | 0.09 (0.2) | 0.08 (0.2) | 0.924 |
| Duration of zinc supplementation (days) (IQR) | 5 (11.0) | 5 (14.0) | 2.5 (10.0) | 0.561 |
| Previous treatment N (%) | | | | |
| • Chemotherapy | 16 (53.3) | 9 (50.0) | 7 (58.3) | 0.654 |
| • Steroid | 7 (23.3) | 5 (27.8) | 2 (16.7) | 0.669 |
| • Blood transfusion | 26 (86.7) | 17 (94.4) | 9 (75.0) | 0.274 |
| • Iron chelation | 9 (30.0) | 8 (44.4) | 1 (8.3) | **0.049** |
| Type of HSCT N (%) | | | | |
| • Haploidentical HSCT | 15 (50.0) | 9 (50.0) | 6 (50.0) | 1.000 |
| • Non-Haploidentical HSCT | 15 (50.0) | 9 (50.0) | 6 (50.0) | |
| Serology N (%) | | | | |
| • Recipient CMV IgG positive | 24 (80.0) | 15 (83.3) | 9 (75.0) | 0.660 |
| • Recipient EBV IgG positive | 20 (66.7) | 11 (61.1) | 9 (75.0) | 0.694 |
| • Donor CMV IgG positive | 22 (84.6) | 12 (66.7) | 10 (83.3) | 0.136 |
| • Donor EBV IgG positive | 20 (76.9) | 11 (61.1) | 9 (75.0) | 0.352 |
| Hematological parameters before HSCT | | | | |
| • Absolute neutrophil count (cells/mm$^3$) (IQR) | 2,788 (2,096.4) | 2,815 (2,248.0) | 2,032 (3,039.0) | 0.553 |
| • Absolute lymphocyte count (cells/mm$^3$) (IQR) | 778.4 (471.9) | 602 (713.0) | 1,163.5 (964.0) | **0.038** |
| • Aspartate transaminase (U/L) (IQR) | 33 (39.0) | 32.5 (17.0) | 39 (53.0) | 0.385 |
| • Alanine aminotransferase (U/L) (IQR) | 41 (51.0) | 42.5 (52.0) | 30.5 (33.0) | 0.597 |
| • Serum albumin (g/L) (mean ± SD) | 40.2 ± 3.4 | 41.3 ± 2.7 | 39.3 ± 3.3 | 0.094 |
| • Creatinine clearance (mL/min) (mean ± SD) | 140.2 ± 40.2 | 135.4 ± 39.2 | 133.5 ± 45.7 | 0.553 |
| Infection status before HSCT N (%) | | | | |
| • Possible or probable invasive pulmonary aspergillosis | 5 (16.7) | 2 (11.1) | 3 (25) | 0.247 |
| • Latent tuberculosis infection | 2 (6.7) | 0 (0) | 2 (16.7) | |
| • Chronic HBV infection | 2 (6.7) | 1 (5.6) | 1 (8.3) | |

[*] The category of patients is based on zinc level on admission day.

IQR: interquartile range; HSCT: hematopoietic stem cell transplant; CMV; cytomegalovirus: EBV; Epstein Barr virus; IgG: immunoglobulin G; HBV: hepatitis B virus; NA: not applicable

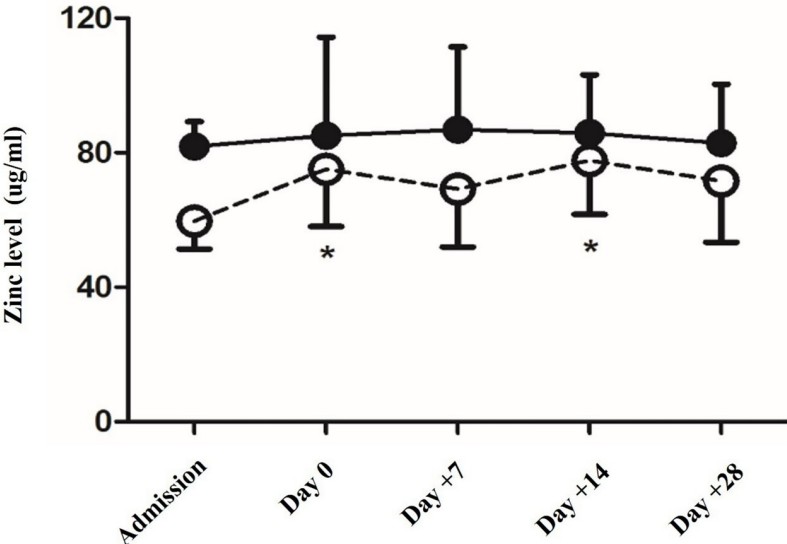

**Fig 1. Plasma zinc levels.** Plasma zinc levels in patients with normal zinc levels at admission (solid circle) and patients with zinc deficiency at admission (open circle) were measured at admission, day 0 (stem cell transfusion), day+7, day +14, and day+28. *Statistically significant difference from baseline (day of admission).

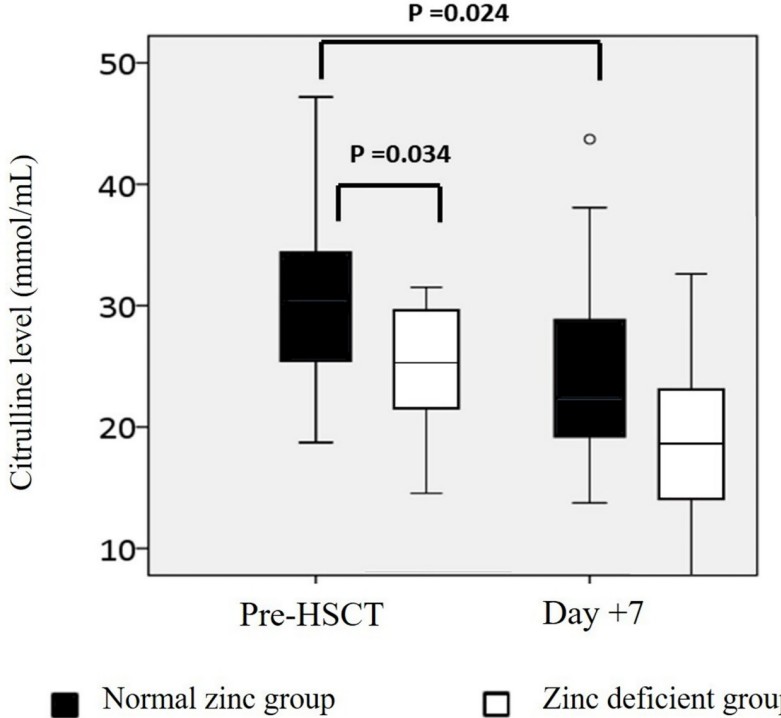

**Fig 2. Plasma citrulline levels.** Plasma citrulline levels were measured in patients with normal zinc levels (black bars) and patients with zinc deficiency (white bars) at admission and day+7.

**Table 2. Non-infectious outcomes after hematopoietic stem cell transplant.**

| Non-infectious outcomes | Total (N = 30) | Normal zinc* (N = 18) | Zinc deficiency* (N = 12) | p-value |
|---|---|---|---|---|
| Febrile episode within 48 hours after HSCT N (%) | 19 (63.3) | 13 (72.0) | 6 (50.0) | 0.266 |
| Maximum temperature of fever within 48 hours, Median (IQR) | 38.6 (1.0) | 38.5 (1.6) | 38.6 (0.6) | 0.224 |
| Day of WBC engraftment, Median (IQR) | 14 (5.0) | 14 (4.0) | 15 (9.0) | 0.691 |
| Day of platelet engraftment, Median (IQR) | 25 (58.0) | 18 (55.0) | 25 (23.0) | 0.820 |
| Primary graft failure N (%) | 2 (6.7) | 1 (5.6) | 1 (8.3) | 1.000 |
| Engraftment syndrome N (%) | 5 (16.7) | 3 (16.7) | 2 (16.7) | 1.000 |
| Mucositis N (%) | 17 (56.7) | 11 (61.1) | 6 (50) | 0.711 |
| VOD N (%) | 7 (23.3) | 5 (27.8) | 2 (16.7) | 0.669 |
| Acute GVHD N (%) | 12 (40.0) | 7 (38.9) | 5 (41.7) | 1.000 |
| Severe (grade 3 and 4) N (%) | 4 (13.3) | 1 (5.56) | 3 (25.0) | 0.222 |
| ICU visit N (%) | 9 (30.0) | 6 (33.0) | 3 (25.0) | 0.704 |
| Dead N (%) | 4 (13.3) | 2 (11.0) | 2 (16.7) | 1.000 |

* The category of patients is based on zinc level on admission day.

HSCT: hematopoietic stem cell transplant; IQR: interquartile range; WBC: white blood count; VOD: veno-occlusive disease; GVHD: graft-versus-host disease; ICU: intensive care unit

after adjusting for age (Table 5). Comparing patients with bacterial infection and those without, citrulline levels at admission and day+7 tended to be lower in patients with bacterial infection but were not significantly different (29.10 ± 7.43 vs 25.58 ± 5.64 mmol/L; $p = 0.260$ at admission and 24.25 ± 9.07 vs 18.94 ± 7.82 mmol/L; $p = 0.176$ at day+7). Characteristics of patients with bacterial infection is summarized in Table 6.

## Discussion

Zinc plays an essential role in regulating immune responses and promoting epithelial integrity. Zinc deficiency leads to impaired barrier function and cellular immunity [2, 24]. Patients undergoing HSCT are at a higher risk for infection and epithelial barrier dysfunction following chemotherapy [13, 25, 26]. Our study showed that the prevalence of zinc deficiency in pediatric patients who were candidates for HSCT was 40%. Zinc-deficient patients also had lower citrulline levels reflecting reduced intestinal epithelial integrity and function. Patients with zinc

**Table 3. Infectious outcomes after hematopoietic stem cell transplant.**

| Infectious outcomes | Total (N = 30) | Normal zinc group* (N = 18) | Zinc deficiency group* (N = 12) | p-value |
|---|---|---|---|---|
| Febrile neutropenia N (%) | 24 (80.0) | 15 (83.3) | 9 (75.0) | 0.660 |
| Viral infection N (%) | 21 (70.0) | 13 (72.2) | 8 (66.7) | 1.000 |
| Viral hemorrhagic cystitis N (%) | 20 (66.7) | 12 (70.6) | 8 (66.7) | 0.419 |
| Fungal infection N (%) | 4 (13.3) | 3 (16.7) | 1 (8.3) | 0.632 |
| Bacterial infection N (%) | | | | |
| Within 14 days | 3 (10.0) | 1 (5.6) | 2 (16.7) | 0.548 |
| Within 90 days | 7 (23.3) | 1 (5.6) | 6 (50.0) | **0.009** |
| Bacteremia N (%) | | | | |
| Within 14 days | 2 (6.7) | 0 (0.0) | 2 (16.7) | 0.152 |
| Within 90 days | 5 (16.7) | 0 (0.0) | 5 (41.7) | **0.006** |

If not specified, infectious outcomes were measured within 90 days after hematopoietic stem cell transplant.

* The category of patients is based on zinc level on admission day.

**Table 4. Risk factors of bacterial infection after hematopoietic stem cell transplant.**

|  | Total (N = 30) | No bacterial infection (N = 23) | Bacterial infection (N = 7) | *p*-value |
|---|---|---|---|---|
| Age (years), (IQR) | 10.6 (9.8) | 10.6 (8.4) | 6.4 (10.0) | **0.039** |
| Sex N (%) |  |  |  |  |
| • Male | 26 (86.7) | 13 (56.5) | 3 (42.9) | 0.675 |
| • Female | 14 (46.7) | 10 (43.5) | 4 (57.1) |  |
| Primary disease N (%) |  |  |  |  |
| • Malignancy | 14 (46.7) | 11 (47.8) | 3 (42.9) | 1.000 |
| • Non-malignancy | 16 (53.3) | 12 (52.2) | 4 (57.1) |  |
| Duration of primary disease before HSCT (years) (IQR) | 4.98 (10.0) | 6.7 (10.0) | 1.3 (4.0) | 0.077 |
| Height for age Z score N (%) |  |  |  |  |
| • Normal | 25 (83.3) | 19 (82.6) | 6 (85.7) | 1.000 |
| • Stunting | 5 (16.7) | 4 (17.4) | 1 (14.3) |  |
| Weight for age Z score N (%) |  |  |  |  |
| • Normal | 30 (100) | 23 (100) | 7 (100) | NA |
| • Underweight | 0 (0) | 0 (0) | 0 (0) |  |
| Previous treatment with iron chelation | 9 (30.0) | 8 (34.8) | 1 (14.3) | 0.393 |
| Type of HSCT N (%) |  |  |  |  |
| • Haploidentical HSCT | 15 (50.0) | 10 (43.5) | 5 (71.4) | 0.390 |
| • Non-Haploidentical HSCT | 15 (50.0) | 13 (56.5) | 2 (28.6) |  |
| ATG in pre-conditioning regimen N (%) | 22 (73.3) | 16 (69.6) | 6 (85.7) | 0.638 |
| Zinc deficiency before HSCT N (%) | 12 (40.0) | 6 (26.1) | 6 (85.7) | **0.009** |
| Zinc level before HSCT (mean ± SD) | 73.0 ± 13.4 | 75.3 ± 11.4 | 65.1 ± 17.0 | **0.042** |
| Fever after stem cell infusion within 48 hours N (%) | 19 (63.3) | 15 (65.2) | 4 (57.1) | 1.000 |
| Engraftment syndrome N (%) | 5 (16.7) | 3 (13.0) | 2 (28.6) | 0.565 |
| Mucositis N (%) | 17 (56.7) | 12 (52.2) | 5 (71.4) | 0.427 |
| Acute GVHD N (%) | 12 (40.0) | 9 (39.1) | 3 (42.9) | 1.000 |
| Severe acute GVHD (grade 3 and 4) N (%) | 4 (13.3) | 3 (13.0) | 1 (14.3) | 1.000 |

deficiency at admission had a higher probability of having microbiologically confirmed bacterial infection compared with those with normal zinc levels.

Few studies have reported plasma zinc levels in patients who underwent HSCT [14, 27, 28]. The prevalence of zinc deficiency in pediatric patients who underwent HSCT was ~34% in one study in Turkey [14], similar to the 40% in our study. The zinc level was the lowest at admission and then increased, which is in contrast to previous studies showing that the lowest zinc level occurred on day+7 and day+15 [5]. Previously, our group also reported an increase in serum retinol levels in pediatric patients after stem cell infusion [29]. This potentially indicates that patients' nutritional status may be improved while they stayed in the hospital.

Zinc-deficient patients were younger and had a shorter duration between the diagnosis of primary disease and the time of HSCT. Several studies have shown that young children were more vulnerable to zinc deficiency [30, 31]. Low intake of zinc-enriched food and increased

**Table 5. Risk factors of bacterial infection after hematopoietic stem cell transplant.**

|  | Unadjusted | | Adjusted | |
|---|---|---|---|---|
|  | Odds ratio | *p*-value | Odds ratio | *p*-value |
| Age | 0.836 (0.687–1.018) | 0.75 | 0.92 (0.75–1.128) | 0.423 |
| Zinc deficiency before HSCT | 17.0 (1.683–171.70) | **0.016** | 12.31 (1.084–139.92) | **0.043** |

**Table 6. Summary of cases with bacterial infection within 90 days after hematopoietic stem cell transplant.**

| N | Age (Year) | Primary disease | Site of infection | Pathogen | Days of bacterial infection (days after HSCT) | Zinc before HSCT (ug/dL) | Zinc at day 0 (ug/dL) | Zinc at day+28 (ug/dL) | Citrulline before HSCT (nmol/mL) | Citrulline at day+7 (nmol/mL) |
|---|---|---|---|---|---|---|---|---|---|---|
| 1 | 1.0 | Hemoglobin Bart disease | Bacteremia (CRBSI) | *S. aureus* | 88 | 68 | 88 | 93 | 24.62 | 15.77 |
| 2 | 1.6 | Osteopetrosis | Pneumonia | *Legionella spp.* | 36 | 66 | 55 | 41 | 14.48 | 6.03 |
| 3 | 6.4 | Bone marrow failure | Bacteremia (CRBSI) | *E. cloacae* | 8 | 58 | 96 | 74 | 27.71 | 22.83 |
| 4 | 7.3 | Neuroblastoma | Pneumonia Bacteremia (CRBSI) with UTI | *E. coli* *E. coli* CRE | 68 71 | 61 | 71 | 70 | 31.81 | 23.14 |
| 5 | 5.8 | AML | Bacteremia | *B. cereus* | 11 | 60 | 61 | 86 | 30.32 | 30.69 |
| 6 | 13.0 | β-Thalassemia | Bacteremia | *S. hemolyticus* | 38 | 43 | 76 | 90 | 25.48 | 14.58 |
| 7 | 11.6 | ALL | Perianal abscess | *E. coli* ESBL | 14 | 100 | 74 | 95 | 24.65 | 19.54 |

CRBSI: catheter related blood stream infection; AML: Acute myeloid leukemia; ALL: Acute lymphoblastic leukemia; UTI; urinary tract infection; CRE: carbapenem resistance enterobacteriaceae; ESBL: extended spectrum beta lactamase

utilization of zinc for growth and development [32] may explain why younger children had lower zinc levels. Poor oral intake during the illness might aggravate zinc deficiency in these patients. However, the patients in this study were not clinically malnourished, and no patients exhibited apparent signs of macro- or micro-nutrient deficiency at admission. In addition, weight and height for age Z-scores were not different between patients with normal zinc levels and those with zinc deficiency. The albumin level between the two groups of patients was also not different. Therefore, these findings emphasize that normal weight and height may not necessarily indicate that the patient does not have subclinical zinc deficiency.

Interestingly, we found that patients who underwent iron chelation therapy tended to have higher zinc levels than those who did not receive this treatment. Iron chelation therapy is well known to be associated with zinc deficiency [33–35]. This may be explained by the fact that patients who did not undergo iron chelation therapy mainly included patients of a younger age, which is associated with zinc deficiency, as described above.

Citrulline is used as an indirect marker for intestinal mucosal integrity [15–17]. We found that patients with zinc deficiency before HSCT also had lower citrulline levels before HSCT. This supports the hypothesis that zinc contributes to intact intestinal mucosal integrity [8, 36]. However, we did not determine other factors that may also affect intestinal mucosal integrity. Additional parameters, such as vitamin or other mineral deficiency, may also contribute to this outcome [37, 38]. In addition, zinc deficiency may arise from impaired intestinal integrity, leading to poor zinc absorption [39, 40]. Previously, we reported a decrease in citrulline levels after stem cell infusion [29]. The results of decrease in citrulline levels after stem cell infusion in this study were consistent with previous findings. This study implied that chemotherapy during conditioning regimens made patients more vulnerable to impaired intestinal epithelial integrity after HSCT. One important caveat is that citrulline is an indirect marker of intestinal integrity. The sensitivity and specificity of using the cut-off level of 20 mmol/L to determine the function of the intestine were 80 and 84%, respectively [16]. Therefore, citrulline level may not be correlated with intestinal function or integrity in every case.

This study showed that patients with zinc deficiency at admission had a higher probability of developing bacterial infection after HSCT, even after adjusting for age. Interestingly, pathogens

that cause bacterial infection were predominately from the skin and gastrointestinal tract, where mucosal barriers and immunity play an essential role in protection against these pathogens [41]. Nearly all patients with bacterial infection had low zinc levels before admission, but the level increased to normal values after admission. These patients might have had chronic zinc deficiency, and the effects on immune function may persist even after the zinc level has normalized. Zinc deficiency leads to decreased thymulin activity and impaired T-cell function [42, 43]. A previous study showed that the zinc level in zinc-deficient patients improved within days, whereas normal thymulin activity was restored after 1 month of zinc supplementation [43]. However, bacterial infection was not significantly associated with low citrulline levels in this study. One potential explanation is that zinc deficiency has deleterious effects on both intestinal integrity and host immune function, predisposing patients to bacterial infections, but impaired intestinal integrity alone did not significantly contribute to the increased incidence of bacterial infection in these patients. In addition, infection could also come from other portals apart from gastrointestinal tract. The finding that zinc deficiency was associated with subsequent bacterial infection should be further verified in patients with zinc deficiency who receive zinc supplementation before HSCT. If zinc supplementation in these patients improves outcomes in terms of bacterial infection, this will support that zinc protects against bacterial infection after HSCT.

The limitations in this study include its small sample size, which may prevent the identification of other potential factors associated with bacterial infection in these patients. Other factors can contribute to bacterial infection after HSCT [44, 45]. Other vitamins or minerals also contribute to integrity of immune system [46–48]. Patients with zinc deficiency may also have concomitant other mineral or vitamin deficiencies culminating in predisposing patients to worse infectious outcomes. To reduce the incidence of bacterial infection in this group of patients, all factors associated with bacterial infection need to be addressed. Another limitation was that a nutritional diary or in-depth history of nutrition was not included in this study to confirm patients' zinc status. In addition, other markers of zinc status, such as tissue zinc concentrations or urinary zinc excretion, were not included.

## Conclusion

The prevalence of zinc deficiency in pediatric patients undergoing HSCT was high. The zinc level was lower in patients with younger age and a shorter duration between the diagnosis of primary disease and the time of HSCT. This pilot study showed that zinc deficiency seemed to be associated with impaired intestinal epithelial integrity reflected by lower citrulline levels. Zinc deficiency tended to increase the risk of bacterial infection within 90 days after HSCT. Further study with larger sample size and including other factors which may be associated with bacterial infections in these patients should be performed to confirm these preliminary findings. Screening of other minerals and vitamins in these patients may lead to clearer picture of how these micronutrients play role in infectious outcomes in these patients. Appropriate zinc or other micronutrient supplementation before HSCT may reduce the incidence of bacterial infection after HSCT.

## Supporting information

**S1 File.**
(DOCX)

**S2 File.**
(DOCX)

**S1 Data.**
(XLSX)

## Acknowledgments

We would like to thank the Department of Toxicology, Ramathibodi Hospital for performing the zinc level analysis. We thank Melissa Crawford, PhD, from Edanz (https://jp.edanz.com/ac) for editing a draft of this manuscript.

## Author Contributions

**Conceptualization:** Warangkhana Suwanphoerung, Samart Pakakasama, Usanarat Anurathapan, Suradej Hongeng, Nalinee Chongviriyaphan, Nopporn Apiwattanakul.

**Data curation:** Usanarat Anurathapan.

**Formal analysis:** Chompunuch Klinmalai, Sasivimol Rattanasiri, Nopporn Apiwattanakul.

**Investigation:** Warangkhana Suwanphoerung.

**Methodology:** Warangkhana Suwanphoerung, Chompunuch Klinmalai, Sasivimol Rattanasiri.

**Supervision:** Suradej Hongeng.

**Writing – original draft:** Warangkhana Suwanphoerung, Nopporn Apiwattanakul.

**Writing – review & editing:** Samart Pakakasama, Usanarat Anurathapan, Nalinee Chongviriyaphan, Nopporn Apiwattanakul.

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
