## [Decision Letter · Decision Letter 0]

29 Sep 2022

PONE-D-22-07042Association of zinc deficiency with infectious complications in pediatric hematopoietic stem cell transplantation patientsPLOS ONE

Dear Dr. Apiwattanakul,

Thank you for submitting your manuscript to PLOS ONE. After careful consideration, we feel that it has merit but does not fully meet PLOS ONE’s publication criteria as it currently stands. Therefore, we invite you to submit a revised version of the manuscript that addresses the points raised during the review process.

Please address the Reviewer 1 and 2's concerns and issues point by point, especially the questions raised by Reviewer 2. ==============================

We look forward to receiving your revised manuscript.

Kind regards,

Jianhua Yu, Ph.D.

Academic Editor

PLOS ONE

Journal Requirements:

3. We note that your study involved tissue/organ transplantation. Please provide the following information regarding tissue/organ donors for transplantation cases analyzed in your study.

a. Please provide the source(s) of the transplanted tissue/organs used in the study, including the institution name and a non-identifying description of the donor(s).

b. Please state in your response letter and ethics statement whether the transplant cases for this study involved any vulnerable populations; for example, tissue/organs from prisoners, subjects with reduced mental capacity due to illness or age, or minors.

- If a vulnerable population was used, please describe the population, justify the decision to use tissue/organ donations from this group, and clearly describe what measures were taken in the informed consent procedure to assure protection of the vulnerable group and avoid coercion.

- If a vulnerable population was not used, please state in your ethics statement, “None of the transplant donors was from a vulnerable population and all donors or next of kin provided written informed consent that was freely given.”

4. In the Methods, please provide detailed information about the procedure by which informed consent was obtained from organ/tissue donors or their next of kin. In addition, please provide a blank example of the form used to obtain consent from donors, and an English translation if the original is in a different language.

5. Please indicate whether the donors were previously registered as organ donors. If tissues/organs were obtained from deceased donors or cadavers, please provide details as to the donors’ cause(s) of death.

6. Please provide the participant recruitment dates and the period during which transplant procedures were done (as month and year).

7. Please discuss whether medical costs were covered or other cash payments were provided to the family of the donor. If so, please specify the value of this support (in local currency and equivalent to U.S. dollars)."""f you are reporting a retrospective study of medical records or archived samples, please ensure that you have discussed whether all data were fully anonymized before you accessed them and/or whether the IRB or ethics committee waived the requirement for informed consent. If patients provided informed written consent to have data from their medical records used in research, please include this information.

8. We note that the grant information you provided in the ‘Funding Information’ and ‘Financial Disclosure’ sections do not match.

Reviewers' comments:

Reviewer's Responses to Questions

**Comments to the Author**

1. Is the manuscript technically sound, and do the data support the conclusions?

Reviewer #1: Partly

Reviewer #2: No

2. Has the statistical analysis been performed appropriately and rigorously? 

Reviewer #1: I Don't Know

Reviewer #2: No

3. Have the authors made all data underlying the findings in their manuscript fully available?

Reviewer #1: No

Reviewer #2: Yes

4. Is the manuscript presented in an intelligible fashion and written in standard English?

Reviewer #1: Yes

Reviewer #2: Yes

5. Review Comments to the Author

Reviewer #1: This is an interesting study which focused on the association of nutrition status with post hematopoietic stem cell transplantation infections in pediatric patients.

However, this study only detected the serum zinc level, but not copper, or iron which also serve as the essential trace elements for the growth, development and infection defense. The results showed in this manuscript may with some bias.

The authors included citrulline as the indicator for enterocyte function which reflected the gastrointestinal barrier integrity. But the infection may come from skin, urinary tract, respiratory tract, and etc. If the authors can detect the serum LPS level which shows the overall inflammation level. The according results may work together with citrulline.

Reviewer #2: 1. Sample size is too small;

2. There are too many interfering factors in the infection of patients undergoing hematopoietic stem cell transplantation, and the conclusions obtained in this paper are not reliable;

3. The conclusion that citrulline represents intestinal integrity is controversial.

6. PLOS authors have the option to publish the peer review history of their article (what does this mean?). If published, this will include your full peer review and any attached files.

Reviewer #1: No

Reviewer #2: No

---

## [Author Response · Author response to Decision Letter 0]

12 Oct 2022

Response to the reviewers

Journal Requirements:

We would like to thank for this suggestion. We corrected accordingly.

We added more details regarding the consent process and re-wrote the sentence as in the following sentence in the last paragraph of “Study Design and Patients”,

Patients or legally authorized representatives gave written informed consent. Patients aged 7-18 years old also gave written assent form.

3. We note that your study involved tissue/organ transplantation. Please provide the following information regarding tissue/organ donors for transplantation cases analyzed in your study.

a. Please provide the source(s) of the transplanted tissue/organs used in the study, including the institution name and a non-identifying description of the donor(s).

b. Please state in your response letter and ethics statement whether the transplant cases for this study involved any vulnerable populations; for example, tissue/organs from prisoners, subjects with reduced mental capacity due to illness or age, or minors.

- If a vulnerable population was used, please describe the population, justify the decision to use tissue/organ donations from this group, and clearly describe what measures were taken in the informed consent procedure to assure protection of the vulnerable group and avoid coercion.

- If a vulnerable population was not used, please state in your ethics statement, “None of the transplant donors was from a vulnerable population and all donors or next of kin provided written informed consent that was freely given.” 

 We added a paragraph giving the detail in this regard,

 “Patients who received haploidentical stem cell transplantation received stem cells from their father or mother. Those who received matched related donor stem cell transplant received stem cells from their siblings and those who received matched unrelated donor stem cell transplant received stem cells from Thai Red Cross. Thai Red Cross is a non-profitable organization which provides stem cells from registered donors for patients who need stem cell transplant but do not have matched related donors. All donors or legally authorized representatives in haploidentical and matched related stem cell transplant provided written informed consent that was freely given. Cash payments or any incentives were not offered to the family of the donors.”

4. In the Methods, please provide detailed information about the procedure by which informed consent was obtained from organ/tissue donors or their next of kin. In addition, please provide a blank example of the form used to obtain consent from donors, and an English translation if the original is in a different language.

 We added this sentence in the 3rd paragraph of “Study design and patients” part as the following,

“All donors or legally authorized representatives in haploidentical and matched related stem cell transplant provided written informed consent that was freely given.”

Blank examples of the forms used to obtain consent from donors, and an English translation are provided in the supporting information.

5. Please indicate whether the donors were previously registered as organ donors. If tissues/organs were obtained from deceased donors or cadavers, please provide details as to the donors’ cause(s) of death.

 In matched unrelated donor stem cell transplant, donors were registered donors at Thai Red Cross as mentioned in this sentence in the third paragraph of “Study Design and Patients”,

 Thai Red Cross is a non-profitable organization which manages to provide stem cells from registered donors for patients who need stem cell transplant but do not have matched related donors.

6. Please provide the participant recruitment dates and the period during which transplant procedures were done (as month and year).

 The dates of the participant recruitment were provided in the “Study design and patients” part, 

“Pediatric patients aged 0–18 years old who underwent HSCT at Ramathibodi Hospital in Thailand during March 2020–September 2021 were enrolled in this study.”

7. Please discuss whether medical costs were covered or other cash payments were provided to the family of the donor. If so, please specify the value of this support (in local currency and equivalent to U.S. dollars). If you are reporting a retrospective study of medical records or archived samples, please ensure that you have discussed whether all data were fully anonymized before you accessed them and/or whether the IRB or ethics committee waived the requirement for informed consent. If patients provided informed written consent to have data from their medical records used in research, please include this information. 

 Cash payments or any incentives were not provided to the family of the donors. We added this sentence in the third paragraph of “Study Design and Patients”.

8. We note that the grant information you provided in the ‘Funding Information’ and ‘Financial Disclosure’ sections do not match.

 We corrected accordingly.

Reviewers' comments:

Reviewer's Responses to Questions

Comments to the Author

1. Is the manuscript technically sound, and do the data support the conclusions?

Reviewer #1: Partly

Reviewer #2: No

2. Has the statistical analysis been performed appropriately and rigorously?

Reviewer #1: I Don't Know

Reviewer #2: No

3. Have the authors made all data underlying the findings in their manuscript fully available?

Reviewer #1: No

Reviewer #2: Yes

4. Is the manuscript presented in an intelligible fashion and written in standard English?

Reviewer #1: Yes

Reviewer #2: Yes

5. Review Comments to the Author

Reviewer #1: This is an interesting study which focused on the association of nutrition status with post hematopoietic stem cell transplantation infections in pediatric patients.

However, this study only detected the serum zinc level, but not copper, or iron which also serve as the essential trace elements for the growth, development and infection defense. The results showed in this manuscript may with some bias.

We would like to thank the reviewer for this important comment. We totally agree that many micronutrients contribute to integrity of immune system. Since zinc is potentially essential in both immune system and epithelial integrity and zinc deficiency is postulated to be common in this group of patients due to poor intake and increased loss by stool, we would like to focus on this element first. We added a sentence with references to emphasize that there are other micronutrients which can also contribute to integrity of immune system in the last paragraph of the “Discussion” part;

“Other vitamins or minerals also contribute to integrity of immune system [46-48]. Patients with zinc deficiency may also have concomitant other mineral or vitamin deficiencies culminating in predisposing patients to worse infectious outcomes.”

We also modified the last part of the “Conclusion” part as the following,

“Screening of other minerals and vitamins in these patients may lead to clearer picture of how these micronutrients play role in infectious outcomes in these patients. Appropriate zinc or other micronutrient supplementation before HSCT may reduce the incidence of bacterial infection after HSCT.”

The authors included citrulline as the indicator for enterocyte function which reflected the gastrointestinal barrier integrity. But the infection may come from skin, urinary tract, respiratory tract, and etc. If the authors can detect the serum LPS level which shows the overall inflammation level. The according results may work together with citrulline.

We appreciate this comment and agree that infection can come from other portals. Lipopolysaccharide level in serum would reflect loss of intestinal integrity as the reviewer commented. However, this is beyond the scope of this study but it is really worth for further study. In this study, low level of citrulline was not associated with infection which would reflect what the reviewer mentioned that infection could come from other portals. We addressed this in the 6th paragraph of the “Discussion” part,

“One potential explanation is that zinc deficiency has deleterious effects on both intestinal integrity and host immune function, predisposing patients to bacterial infections, but impaired intestinal integrity alone did not significantly contribute to the increased incidence of bacterial infection in these patients. In addition, infection could also come from other portals apart from gastrointestinal tract.”

We also modified the conclusion part of the abstract to mention that citrulline level was not different between patients with and without bacterial infections.

“Zinc-deficient patients had lower citrulline levels and higher incidence of bacterial infection after HSCT. However, citrulline level was not different between patients with and without bacterial infections. It is worth to investigate whether zinc supplementation before HSCT can reduce the incidence of bacterial infection after HSCT.”

Reviewer #2: 1. Sample size is too small;

2. There are too many interfering factors in the infection of patients undergoing hematopoietic stem cell transplantation, and the conclusions obtained in this paper are not reliable;

We greatly appreciate this comment. We admit that small sample size and the fact that many factors that may contribute to infections in these patients were not measured are main limitations in this study. We toned down our findings by using the highlighted words or phrases in the conclusion part as the following, 

“This pilot study showed that zinc deficiency seemed to be associated with impaired intestinal epithelial integrity reflected by lower citrulline levels. Zinc deficiency tended to increase the risk of bacterial infection within 90 days after HSCT. Further study with larger sample size and including other factors which may be associated with bacterial infections in these patients should be performed to confirm these preliminary findings.”

3. The conclusion that citrulline represents intestinal integrity is controversial.

We would like to thank the reviewer for this comment. Citrulline is considered to be an indirect marker of intestinal integrity, not a direct one. Some factors may interfere the interpretation of citrulline as the marker of intestinal function and enterocyte mass, hence, representing intestinal integrity. There is a systematic review showing that citrulline level could be used to represent intestinal function or enterocyte mass (as in reference 16 of the manuscript; Fragkos K, et al 2018). We modified the first sentence in the 5th paragraph of the “Discussion part as the following,

“Citrulline is used as an indirect marker for intestinal mucosal integrity [15-17].”

In addition, we add sentences regarding the limitation of citrulline in the last part of the 5th paragraph in the “Discussion” part.

“One important caveat is that citrulline is an indirect marker of intestinal integrity. The sensitivity and specificity of using the cut-off level of 20 mmol/L to determine the integrity of the intestine were 80 and 84%, respectively [16] . Therefore, citrulline level may not be correlated with intestinal function or integrity in every case.”

6. PLOS authors have the option to publish the peer review history of their article (what does this mean?). If published, this will include your full peer review and any attached files.

Do you want your identity to be public for this peer review? For information about this choice, including consent withdrawal, please see our Privacy Policy.

Reviewer #1: No

Reviewer #2: No

---

## [Decision Letter · Decision Letter 1]

7 Dec 2022

Association of zinc deficiency with infectious complications in pediatric hematopoietic stem cell transplantation patients

PONE-D-22-07042R1

Dear Dr. Apiwattanakul,

We’re pleased to inform you that your manuscript has been judged scientifically suitable for publication and will be formally accepted for publication once it meets all outstanding technical requirements.

Kind regards,

Jianhua Yu, Ph.D.

Academic Editor

PLOS ONE

Additional Editor Comments (optional):

Reviewers' comments:

Reviewer's Responses to Questions

**Comments to the Author**

1. If the authors have adequately addressed your comments raised in a previous round of review and you feel that this manuscript is now acceptable for publication, you may indicate that here to bypass the “Comments to the Author” section, enter your conflict of interest statement in the “Confidential to Editor” section, and submit your "Accept" recommendation.

Reviewer #1: All comments have been addressed

Reviewer #2: All comments have been addressed

2. Is the manuscript technically sound, and do the data support the conclusions?

Reviewer #1: Partly

Reviewer #2: Yes

3. Has the statistical analysis been performed appropriately and rigorously? 

Reviewer #1: I Don't Know

Reviewer #2: Yes

4. Have the authors made all data underlying the findings in their manuscript fully available?

Reviewer #1: Yes

Reviewer #2: Yes

5. Is the manuscript presented in an intelligible fashion and written in standard English?

Reviewer #1: Yes

Reviewer #2: Yes

6. Review Comments to the Author

Reviewer #1: (No Response)

Reviewer #2: This study is relatively new, and there are few related studies. The questions asked were answered relatively satisfactorily.

7. PLOS authors have the option to publish the peer review history of their article (what does this mean?). If published, this will include your full peer review and any attached files.

Reviewer #1: No

Reviewer #2: No

---

## [Editor Report · Acceptance letter]

15 Dec 2022

PONE-D-22-07042R1 

Association of zinc deficiency with infectious complications in pediatric hematopoietic stem cell transplantation patients 

Dear Dr. Apiwattanakul:

I'm pleased to inform you that your manuscript has been deemed suitable for publication in PLOS ONE. Congratulations! Your manuscript is now with our production department. 

Kind regards, 

on behalf of

Dr. Jianhua Yu 

Academic Editor

PLOS ONE